# Mesenchymal Stem Cell-Derived Extracellular Vesicles for Therapeutic Use and in Bioengineering Applications

Caroline McLaughlin [1], Pallab Datta [2], Yogendra P. Singh [3], Alexis Lo [1], Summer Horchler [1], Irina A. Elcheva [4], Ibrahim T. Ozbolat [3,†], Dino J. Ravnic [1] and Srinivas V. Koduru [1,5,*]

1    Department of Surgery, Penn State Health Milton S. Hershey Medical Center, Hershey, PA 17033, USA
2    Department of Pharmaceutics, National Institute of Pharmaceutical Education and Research (NIPER) Kolkata, West Bengal 700054, India
3    Department of Biomedical Engineering, Materials Research Institute, The Huck Institutes of Life Sciences, Penn State University, University Park, PA 16802, USA
4    Department of Pediatrics, Hematology/Oncology, Penn State College of Medicine, Hershey, PA 17033, USA
5    Department of Cellular and Molecular Physiology, Penn State College of Medicine, Hershey, PA 17033, USA
*    Correspondence: skoduru@psu.edu; Tel.: +1-717-531-4332
†    Sabbatical affiliation: Department of Medical Oncology, Cukurova University, Adana 01330, Turkey.

**Abstract:** Extracellular vesicles (EVs) are small lipid bilayer-delimited particles that are naturally released from cells into body fluids, and therefore can travel and convey regulatory functions in the distal parts of the body. EVs can transmit paracrine signaling by carrying over cytokines, chemokines, growth factors, interleukins (ILs), transcription factors, and nucleic acids such as DNA, mRNAs, microRNAs, piRNAs, lncRNAs, sn/snoRNAs, mtRNAs and circRNAs; these EVs travel to predecided destinations to perform their functions. While mesenchymal stem cells (MSCs) have been shown to improve healing and facilitate treatments of various diseases, the allogenic use of these cells is often accompanied by serious adverse effects after transplantation. MSC-produced EVs are less immunogenic and can serve as an alternative to cellular therapies by transmitting signaling or delivering biomaterials to diseased areas of the body. This review article is focused on understanding the properties of EVs derived from different types of MSCs and MSC–EV-based therapeutic options. The potential of modern technologies such as 3D bioprinting to advance EV-based therapies is also discussed.

**Keywords:** stem cells; EVs; extracellular vesicles; bioprinting; biomarkers; MSCs

## 1. Introduction

Extracellular vesicles (EVs) are extremely small, acellular, lipid-membrane enclosed vesicles that are secreted by cells into the extracellular space to modulate cellular communication and important physiological processes, including angiogenesis [1,2]. These heterogeneous groups of particles are released from a variety of cells during normal, stressed, and diseased conditions to optimize the local milieu for the circumstance [3]. EV size can range from 30 nm to more than 2 μm [4,5] and they are mainly categorized into three major classes based on their biogenesis: exosomes (small EVs or sEVs; 30 nm to 150 nm in size), microvesicles (MVs; 150 nm to 1 μm in size) [1,6], and apoptotic bodies (500 nm to 2 μm) [7] (Figure 1).

EVs function by carrying and releasing a defined cargo payload, which can include various proteins (cytokines, chemokines, growth factors, interleukins (ILs)), transcription factors, and nucleic acids (DNA, mRNAs, microRNAs, piRNAs, LncRNAs, sn/snoRNAs, mtRNAs and circRNAs; Figure 2) [8,9]. EVs are now also considered to play a vital role in cell-to-cell communication and may mediate many physiologic processes, such immunomodulation and angiogenesis. For example, during tissue development, remodeling, and wound repair, proangiogenic EVs can be released from endothelial cells, various

progenitor cells, leukocytes, and platelets [10–14]. While interesting, the potential of mesenchymal stem cells (MSCs) has most profoundly captured the curiosity of the medical and scientific community.

| | Small EVs (sEVs) | Medium EVs (mEVs) | Large EVs (lEVs) |
|---|---|---|---|
| **Size** | <100 nm | <200 nm | >200 nm |
| **Made** | Endocytic | Plasma membrane | Plasma membrane |
| **Markers** | CD81, CD63 and CD9 | Integrins, Selectins and CD40 | Annexin V and Phosphatidylserine |
| **Cargo** | mRNA, non-coding RNAs, cytokines and proteins | mRNA, non-coding RNAs, cytokines and proteins | Nuclear fractions and cell organelles |
| **Function** | Intercellular communication | Intercellular communication | Facilitate Phagocytosis |

**Figure 1.** Main types of extracellular vesicles and their size, source, function, their biomarkers, and carrying cargo.

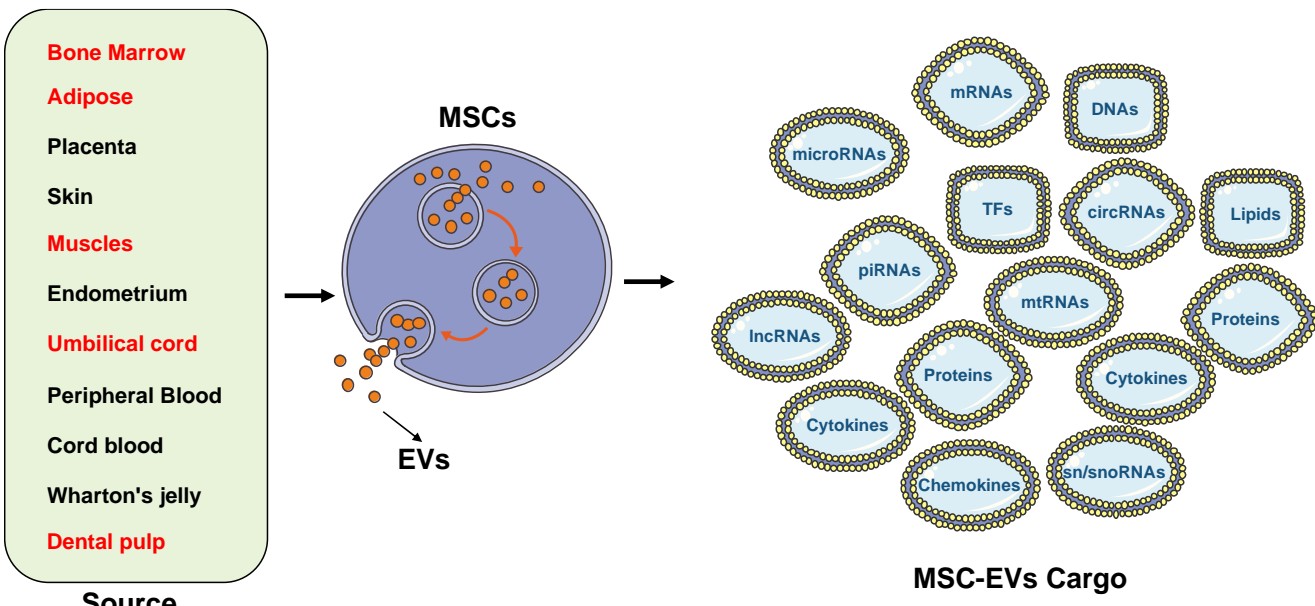

**Figure 2.** Schematic diagram of various sources of MSCs and their EVs that carrying cargo.

It is now known that mesenchymal stem cells (MSC) also secrete a unique EV pool in response to the underlying microenvironment. Recent literature supports MSCs' function through a complex paracrine mechanism, mediated in part by EVs, which facilitate locoregional cell-to-cell communication [15,16]. These MSC-derived heterogeneous EVs possess regenerative potential that can be developed for acellular therapeutics for various diseases (Figure 3) [5,17,18]. MSCs are adult multipotent progenitor cells found in various tissues, such as bone marrow, umbilical cord, and adipose tissue [9]. However, it is still unclear how this affects the corresponding EV functionality.

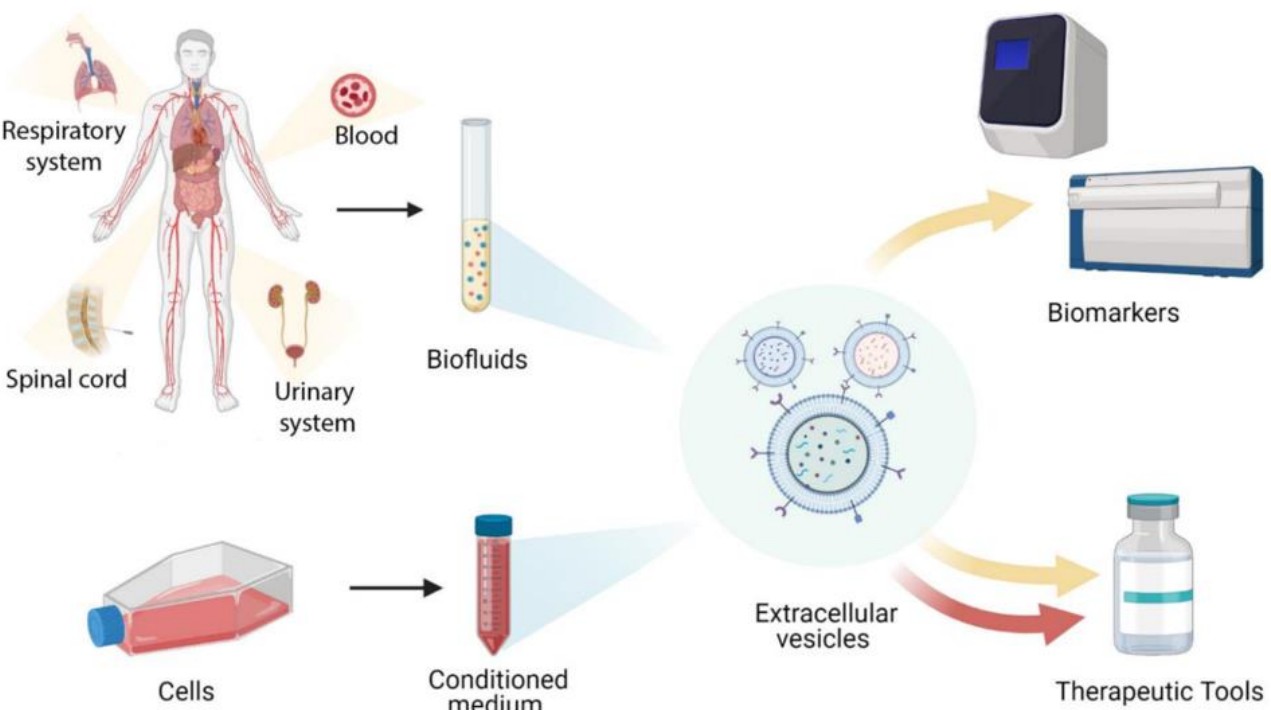

**Figure 3.** Schematic diagram of EV sources and their use in therapeutic and biomarker applications. Reprinted from [18].

In this review, we provide a broad overview of what is known about EVs from specific MSC sources, limited instances of clinical translation, and how they have been applied to emerging bioengineering research.

## 2. Source of Stem Cell Extracellular Vesicles (Figure 4)

### 2.1. Bone Marrow Mesenchymal Stem Cell-Derived EVs

Stem cells derived from bone marrow (BM) have long been utilized clinically to treat various hematopoietic malignancies. However, it has now been shown that even their EVs alone may hold therapeutic application. For example, BM-MSC-EVs have been found to be released under conditions of hypoxia to promote neoangiogenesis in vitro and in vivo [19]. This is consistent with the finding that they increase cancer cell survival under stress and support breast tumor growth in vivo [20]. There is evidence that these EVs regulate osteoblast activity and differentiation in vitro [21] and can even promote bone regeneration in vivo [21]. Others have also shown that BM-MSC-EVs can also enhance tendon healing, likely by modulating macrophage phenotypes and creating an anti-inflammatory environment [22]. BM-MSC-EVs are also thought to have a regenerative role on renal cells [23] and may play a role in attenuating renal fibrosis [24].

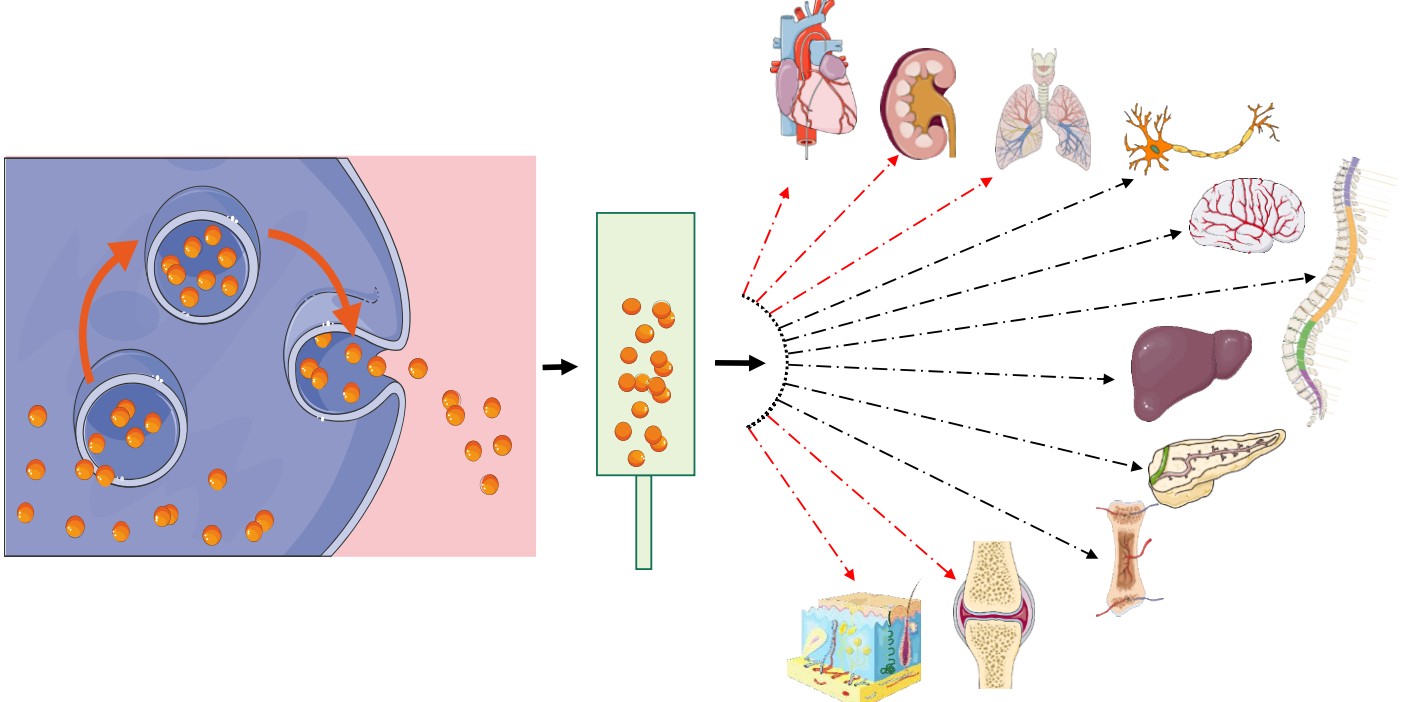

**Figure 4.** Schematic diagram of overall MSC isolation and their clinical applications in various organs and treatments (literature described in manuscript indicated in red arrows).

### 2.2. Adipose Mesenchymal Stem Cell-Derived EVs

Adipose-derived stem cell (ADSC) EVs are another subset found to have multiple biological effects, including angiogenesis, cell survival, apoptosis regulation, inflammation modulation, and tissue regeneration [25]. Although ADSC-EV research is sparse, there is much excitement in the field. This is because ADSC retrieval is technically simple and yields abundant quantities. Furthermore, ADSC-EVs have a higher proliferative rate and lower senescence compared with other types of MSC-EV sourcing [25]. Research studies have shown that ADSC-EVs may have some utility in in the treatment of neurologic diseases by decreasing pathologic proteins in Alzheimer's and Huntington's disease [26,27] and normalizing protein levels in Amyotrophic lateral sclerosis [28]. Other studies have shown a benefit in increasing the amount of muscle and peripheral nerve fibers [29,30], potentially playing a role in hepatic regeneration [31].

### 2.3. Dental Pulp Stem Cells

EVs have also been isolated from dental pulp stem cells (DPSCs). As an MSC source, DPSCs have an excellent proliferative capacity, can produce abundant amounts of EVs, and are thought to have advantages for osteogenic differentiation and decreased cell apoptosis and senescence [32,33]. Dental pulp can be obtained from wisdom teeth, deciduous teeth, supernumerary teeth, and impacted teeth. DPSC-EVs have been shown to mitigate hematopoietic damage after radiation [34] and may play a role in retinal neuroprotection and treatment [35].

### 2.4. Umbilical Cord Mesenchymal Stem Cell-Derived EVs

EVs have been isolated from stem cells of the human umbilical cord (hUC) as well. EV-MSC-hUCs (EV: extracellular vesicle; MSC: mesenchymal stem cells; hUC: human umbilical cord) have been shown to have protective effects in cerebral ischemia/reperfusion injury [36] and may aid in improving alveolarization and angiogenesis in lung injury [37]. Similar to other MSC-EVs, they can promote cancer cell growth in some instances (specifically lung adenocarcinoma) [38] and have antiproliferative effects in other instances (en-

dometrial) [39]. Other studies note the potential of hUC-MSC-EVs (EV: extracellular vesicle; MSC: mesenchymal stem cells; hUC: human umbilical cord) in improving nerve regeneration after transection [40]. EVs also specifically isolated from Wharton's Jelly MSCs have also been found to have a diverse array of functions. For example, those that have been found to suppress CD4-expressing T cells [41] are renally and neurally protective and can decrease oxidative stress [42,43]. They also may play a role in stimulating muscle regeneration [44].

### 2.5. Skeletal Muscle Stem Cells

EVs isolated from skeletal muscle cells have also shown to efficiently promote myogenesis and muscle regeneration [45]. These sEVs contain numerous signals, including myogenic growth factors related to muscle development such as insulin-like growth factor, fibroblast growth factor 2, and platelet-derived growth factor [46]. In vivo studies have demonstrated the effects of these sEVs on reducing damaged areas.

## 3. Cargo: EVs Carry a Wide Range of Materials for Molecular Functions

EVs effectively serve as a signature of tissue type which may be reflected in their plentiful cargo assortment of bioactive proteins, lipids, and coding and noncoding RNAs. Release of this cargo allows for a means of intercellular communication and modification of recipient cells, including functional genetic material exchange [3,47]. Several databases have been created to organize the carrying cargo of EVs, including Vesiclepedia [48], ExoCarta [49], EVpedia [50], exoR BASE [51], and EVmiRNA [52].

### 3.1. RNA

The most well-studied and plentiful content of EVs is mRNA and microRNA (miRNA). miRNAs are 19–24-nucleotide noncoding RNA segments that serve as post-transcriptional regulators of up to 30% of mammalian genes [53–55]. It has been shown that miRNAs are selectively sorted into EVs, secreted through a ceramide-dependent pathway, and taken up by target cells identified by surface receptors and adhesion molecules that facilitate fusion and endocytosis [15]. EVs have the ability to horizontally transfer genetic information to recipient cells, resulting in epigenetic modification [11]. This finding has enlightened realms of diagnostics, as the genetic material in the pathophysiologic EV could serve as a signature, as well as unlock doors for therapeutics to alter pathologic cells with EVs as the vector. Moreover, EV mRNA and miRNA have demonstrated functionality in altering gene expression in recipient cells [11,47,56].

### 3.2. Proteins

Proteins carried in EVs are typically derived from the cytosol, plasma membrane, or endosome; proteins from specific organelles are typically spared from inclusion in the EV [57]. Protein composition in EVs is not dependent on protein quantity in the cell of origin, but rather relies on a regimented protein-sorting mechanism during EV production, though the exact mechanism remains poorly understood [58].

### 3.3. Lipids

Lipid composition of EVs has been published with less frequency compared to protein and nucleic acid composition. Lipid composition of EVs depends on the cell of origin, as their lipid bilayer largely resembles the donor cell plasma membrane, including phospholipids, sphingomyelin, ganglioside, GM3, and cholesterol [58]. Compared to donor cells, EVs tend to be enriched with phosphatidylserine, desaturated phosphatidylethanolamine, desaturated phosphatidylcholine, sphingomyelin, and cholesterol to provide structural support to the vesicle [58]. While thought to be relatively inert, there have been increasing publications on biologic activity of lipid cargo, namely sphingomyelin-mediated tumor angiogenesis [59] and prostaglandin-mediated signaling pathways [60].

## 4. Mesenchymal Stem Cell EVs for Clinical Use

MSC-EVs have been studied extensively in preclinical applications across organ systems [18,61]. They share the advantages of MSCs in terms of derivation from multiple tissue types and potential restoration of a wide variety of damaged tissue and pathophysiologic processes. However, the excitement surrounding MSC-EVs stems from their theoretical advantages over MSCs that have thus far panned out in the literature. First, their intercellular facilitations are paracrine in nature, suggesting that MSC-EVs can mediate the therapeutic effects of MSCs without requiring cellular integration [62]. Second, MSC-EVs do not have the ability to self-replicate, ameliorating the fear of uncontrolled cellular replication that accompanies the use of MSCs. While there is yet to be a recognized and standardized isolation procedure for MSC-EVs [63], the culture process is less invasive, their manufacturing is higher-yield, and the EVs are more stable in that their luminal contents are protected from degradation by their membranes [64]. In addition to pragmatic benefits, multiple studies have demonstrated a superior outcome when comparing MSC-EVs with MSCs [65,66].

Due to their heterogeneity and involvement in a vast amount of physiologic and pathophysiologic processes, MSC-EVs have been investigated in a multitude of tissue types [61]. At the center of their potential lies their ability to mediate immune responses, hemostasis, and angiogenesis [67], particularly in tissue types where endogenous regeneration is ineffective. Indeed, evidence spans across clinical disciplines, demonstrating that EVs mitigate destruction following myocardial ischemia, protect against renal injury, recover lung and liver injuries, and have regenerative potential against nerve injury, largely mediated by their anti-inflammatory, antiapoptotic effects, and proangiogenetic ability [68].

### 4.1. Lung

#### 4.1.1. Lung Injury

MSC-EVs carry specialized cargo that can be used to treat lung diseases. MSC-EVs contains pro/anti-inflammatory properties, reduce oxidative stress, and are remodeled in a variety of in vivo inflammatory lung disease models (Figure 5) [69]. Acute and chronic respiratory disorders are leading cause of the deaths worldwide and chronic lung disease with lowest curative option over a half billion people [70]. Alternative treatments developed in clinics to fight this disease use MSC-EVs, which possess regenerative capacity [71]. Several studies have shown that they enhance acute and chronic conditions in preclinical settings [72–75].

#### 4.1.2. Respiratory Disease (COVID-19)

The recent outbreak of coronavirus disease 2019 (COVID-19) has infected more than 267 million people, with more than 5 million deaths [76]; this devastating illness has made us look for alternative treatments. MSC-derived EVs have the therapeutic potential to treat inflammatory and other diseases (Figure 6) [77]. There have been recent studies of MSC-derived EVs for therapy to mitigate cytokine storms in COVID-19 and promote tissue repair in severely ill patients, as these EVs have immunomodulatory and regenerational capabilities. Chutipongtanate et al. showed that MSC-derived EVs induced COVID-19-infected lung epithelial cells to suppress viral replication and mitigate the production of infectious virions [76]. The SARS-CoV virus infects human cells by recognizing the angiotensin-converting enzyme (ACE2) receptor of host cells [78]. There are anti-inflammatory drugs and a recombinant IL antagonist that can be used to treat this infection; however, these treatments help to improve recovery but not restore lung damage. MSC-EVs from the bone marrow, adipose, and umbilical cord showed improved recovery and survival; several clinical trials are in progress now (Table 1).

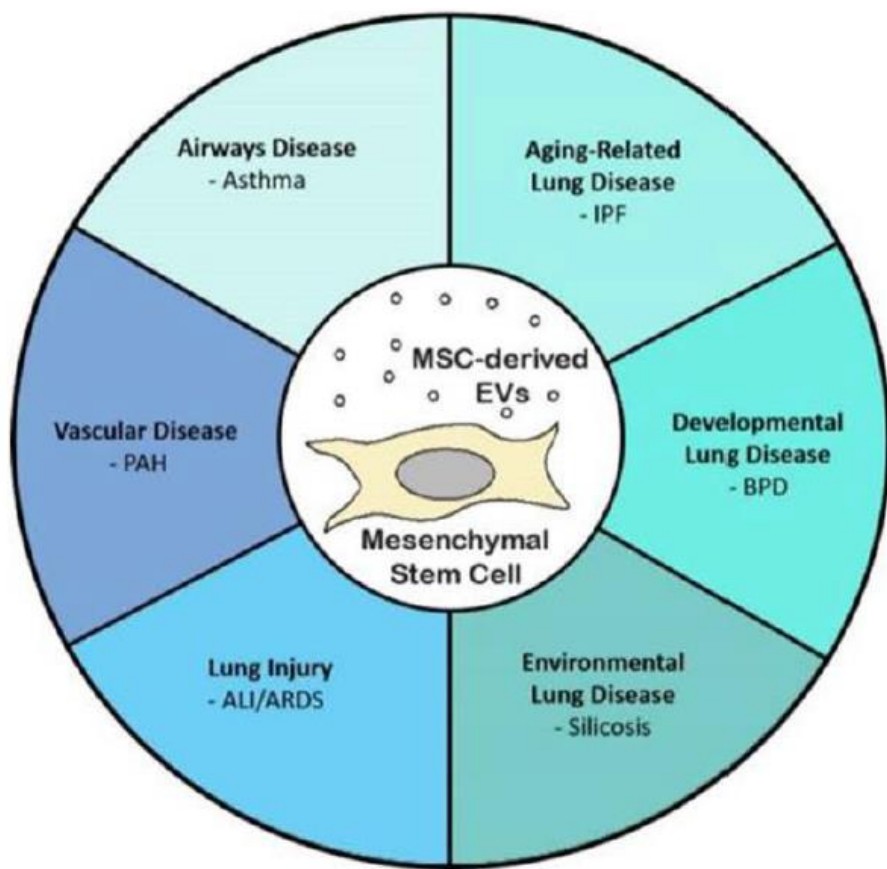

**Figure 5.** MSC-EVs for potential therapeutics for lung diseases. Reprinted from [69].

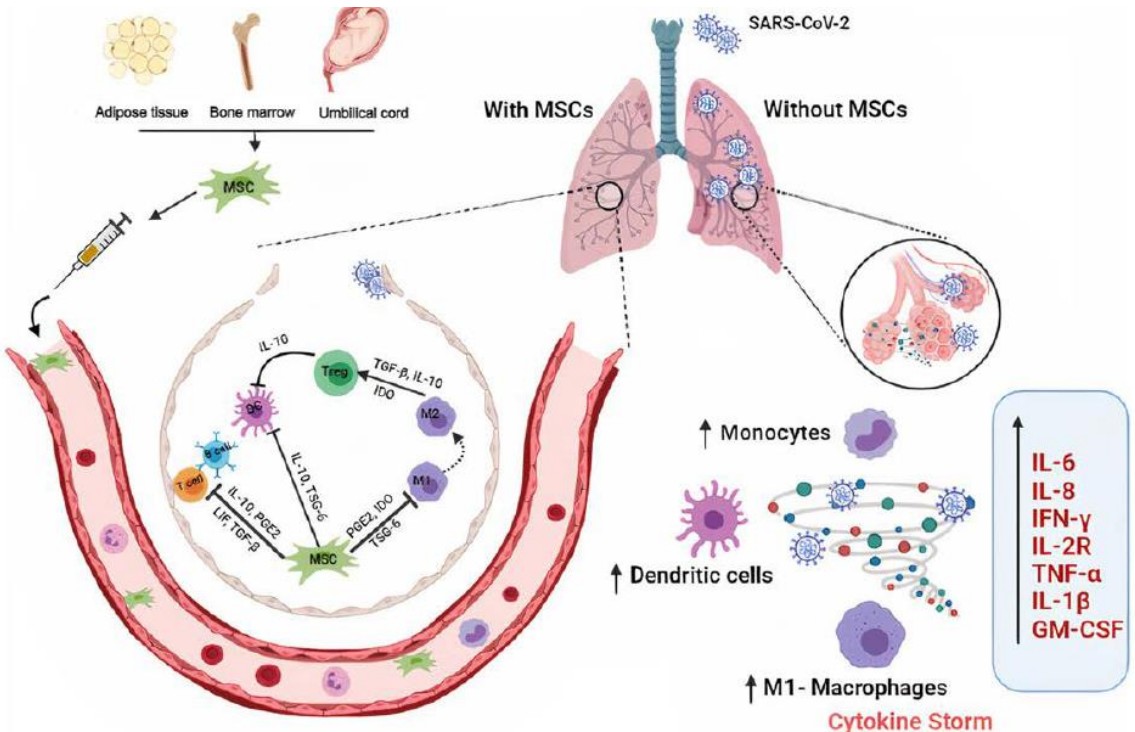

**Figure 6.** Schematic diagram of cytokine storm produced by SARS-CoV2 infection. Reprinted from [77].

**Table 1.** List of registered clinical trials using MSC-EVs. (Source: Clinicaltrials.gov; accessed on 18 March 2022).

| Title | Conditions | Interventions | URL |
|---|---|---|---|
| A Safety Study of IV Stem Cell-derived Extracellular Vesicles (UNEX-42) in Preterm Neonates at High Risk for BPD | Bronchopulmonary Dysplasia | Biological: UNEX-42 | Biological: Phosphate-buffered saline | https://ClinicalTrials.gov/show/NCT03857841 |
| MSC EVs in Dystrophic Epidermolysis Bullosa | Dystrophic Epidermolysis Bullosa | Drug: AGLE 102 | https://ClinicalTrials.gov/show/NCT04173650 |
| Mesenchymal Stem Cell Therapy for SARS-CoV-2-related Acute Respiratory Distress Syndrome | Covid-19 | Biological: Cell therapy protocol 1 | Biological: Cell therapy protocol 2 | https://ClinicalTrials.gov/show/NCT04366063 |
| Extracellular Vesicle Infusion Treatment for COVID-19 Associated ARDS | Covid19 | ARDS | Pneumonia, Viral | Biological: DB-001 | Other: Intravenous normal saline | https://ClinicalTrials.gov/show/NCT04493242 |
| A Global Expanded Access Protocol on Bone Marrow Mesenchymal Stem Cell Derived Extracellular Vesicle Infusion Treatment for Patients With COVID-19 Associated ARDS | Covid19 | ARDS | Hypoxia | Cytokine Storm | Biological: Bone Marrow Mesenchymal Stem Cell Derived Extracellular Vesicles Infusion Treatment | https://ClinicalTrials.gov/show/NCT04657458 |
| ExoFlo™ Infusion for Post-Acute COVID-19 and Chronic Post-COVID-19 Syndrome | Covid19 | Postviral Syndrome | Dyspnea | Biological: Bone Marrow Mesenchymal Stem Cell Derived Extracellular Vesicles | Other: Saline | https://ClinicalTrials.gov/show/NCT05116761 |
| Bone Marrow Mesenchymal Stem Cell Derived Extracellular Vesicles Infusion Treatment for Mild-to-Moderate COVID-19: A Phase II Clinical Trial | COVID-19 | Drug: ExoFlo | https://ClinicalTrials.gov/show/NCT05125562 |
| Bone Marrow Mesenchymal Stem Cell Derived Extracellular Vesicles Infusion Treatment for ARDS | ARDS, Human | Drug: Bone Marrow Mesenchymal Stem Cell Derived Extracellular Vesicles | Other: Saline | https://ClinicalTrials.gov/show/NCT05127122 |
| A Phase I Study of ExoFlo, an ex Vivo Culture-expanded Adult Allogeneic Bone Marrow Mesenchymal Stem Cell Derived Extracellular Vesicle Isolate Product, for the Treatment of Medically Refractory Crohn's Disease | Crohn Disease | IBD - Irritable Bowel Disease | Drug: Bone Marrow MSC Derived Extracellular Vesicle Isolate | https://ClinicalTrials.gov/show/NCT05130983 |
| Intermediate Size Expanded Access for the Use of ExoFlo in the Treatment of Abdominal Solid Organ Transplant Patients Who Are at Risk of Worsening Allograft Function With Conventional Immunosuppressive Therapy Alone | Solid Organ Transplant Rejection | Organ Rejection Transplants | Organ Rejection | Biological: Bone Marrow Mesenchymal Stem Cell Derived Extracellular Vesicles Infusion Treatment | https://ClinicalTrials.gov/show/NCT05215288 |
| Evaluation of Personalized Nutritional Intervention on Wound Healing of Cutaneous Ulcers in Diabetics | Foot, Diabetic | Dietary Supplement: Personalized Nutritional Intervention | https://ClinicalTrials.gov/show/NCT05243368 |

**Table 1.** *Cont.*

| Title | Conditions | Interventions | URL |
|---|---|---|---|
| Bone Marrow Mesenchymal Stem Cell Derived Extracellular Vesicles as Early Goal Directed Therapy for COVID-19 Moderate-to-Severe Acute Respiratory Distress Syndrome (ARDS): A Phase III Clinical Trial | COVID-19 Acute Respiratory Distress Syndrome | Drug: EXOFLO | https://ClinicalTrials.gov/show/NCT05354141 |
| Safety and Effectiveness of Placental Derived Exosomes and Umbilical Cord Mesenchymal Stem Cells in Moderate to Severe Acute Respiratory Distress Syndrome (ARDS) Associated With the Novel Corona Virus Infection (COVID-19) | COVID-19 Acute Respiratory Distress Syndrome\|Respiratory Distress Syndrome | Drug: EV-Pureâ„¢ and WJ-Pureâ„¢\|Drug: Placebo | https://ClinicalTrials.gov/show/NCT05387278 |
| Safety and Efficacy of Injection of Human Placenta Mesenchymal Stem Cells Derived Exosomes for Treatment of Complex Anal Fistula | Fistula Perianal | Other: placenta-MSCs derived exosomes | https://ClinicalTrials.gov/show/NCT05402748 |

## 4.2. Cardiac

Treatment surrounding acute and chronic cardiac ischemia is currently focused on decreased oxygen demand and increasing supply. Regeneration of endogenous cardiomyocytes is an ineffective process and is thought to generate from preexisting cardiomyocytes [79]. As such, once a scar is formed, local destruction and dilation results in irreversible heart failure. Due to the poor ability of endogenous cardiac myocytes to regenerate following ischemic insult, the study of MSC-EVs in cardiac regeneration is particularly promising in regard to its possible clinical benefit following myocardial ischemia. The first study demonstrating decreased cardiac infarct size was published in 2010, wherein Lai et al. utilized a mouse model of ischemia-reperfusion injury, revealing paracrine mediation of angiogenesis and decreased apoptosis through 50–100 nm sEVs [80]. This group went on to reveal increased ATP levels and decreased oxidative stress through the activation of the PI3k/Akt pathway via MSC-EVs in vivo [81]. Since these discoveries, EVs derived from bone marrow- [19], embryonic- [82], umbilical cord- [83], and amniotic fluid [84] MSCs, as well as cardiac progenitor cells [85,86], have demonstrated cardioprotective effects in the context of ischemia. Inhibition of cardiomyocyte apoptosis was found to be partially mediated by the microvesicles' transfer of mi-R221 [87]. sEVs from CD34[+] stem cells and cardiac progenitor cells increased endothelial cell viability, migration, and angiogenesis [88,89]. Further studies on the exact mediators of these effects demonstrate that sEV microRNA-21-5p increased cardiac contractility and calcium handling, likely via PI3K signaling [90].

## 4.3. Renal

The first work describing the emergence of EVs in renal protection was published over a decade ago [91], in which BM-MSC-EVs conferred resistance of tubular cells to apoptosis and induction of tubular cell proliferation. Since this discovery, there has been an increasing number of publications investigating the role of EVs in tubular cell recovery in the context of both acute and chronic kidney injury. The therapeutic modalities for kidney injury are particularly intriguing, given that the current treatments for acute or chronic kidney injury, renal replacement, and dialysis, respectively, are supportive and invasive measures that do not address the underlying insult.

Independent of mechanism or inciting agent causing renal insult, MSC-EVs demonstrate amelioration of acute renal injury in rat and mice models through their proliferative, proangiogenic, antifibrotic, antiapoptotic, and anti-inflammatory effects [92]. These effects are mediated through the horizontal transfer of mRNA [91,93], including human IGF-1 receptor mRNA [94], and miRNA [95,96] from MSC-EVs to injured renal cells. There is also early evidence suggesting that this mode of communication is reciprocal, in that EVs from injured renal cells can communicate with stem cells in the setting of injury [97].

The prediction and early detection of acute kidney injury (AKI) are challenging [98] and current treatments focus on preventing the initial insult and slowing disease progression. Although the kidney has regenerative capacity after acute injury [99], the mortality rate among patients with severe AKI remains high, and only a single episode of AKI predisposes the kidney to the development of chronic kidney disease (CKD) [92,100]. Renal replacement therapy can be used for treatment in the maintenance phase of AKI, but an increase in its dosage does not improve outcomes [101]; thus, the study of MSC-EVs as a treatment modality is necessary. In 2008, for the first time, human BM-MSC was successfully used to prevent AKI induced by cisplatin in mice [102]. Mice injected with BM-MSC showed preserved tubular epithelial and peritubular microvessel integrity, a significant reduction in apoptotic cells, and a prolonged overall survival [102]. In a similar model of cisplatin-induced renal failure, indistinguishable levels of protection were seen after intravenous and intraperitoneal injections of BM-and ADSC-MSCs in adult mice. Isolated MSCs were cultured in conditioned medium and numerous epithelial colonies were found in all tubular regions, suggesting that endogenous tubular cells from all kidney

compartments have the ability to proliferate and migrate [103]. Two recently published meta-analyses pooled the previously published preclinical work on EVs in AKI. The first included 45 studies focused on EVs derived exclusively from MSCs, determining that MSC-EVs produced an increased therapeutic effect on renal recovery compared to MSC-conditioned medium [104]. The second metanalysis analyzed 31 studies, including a variety of EV sources, and found stem cell-derived EVs to be equally effective compared to stem cells in the treatment of AKI. Importantly, the analysis also demonstrated that EV source and delivery dose to have independent effects on the efficacy of EVs [105].

EVs also show emerging promise in chronic kidney disease (CKD) given that their antifibrotic and anti-inflammatory effects that could mitigate ongoing damage. In a rat model of diabetic nephropathy, urine-derived stem cell sEVs prevented the progression of diabetic kidney injury via a reduction in microalbumin secretion, decreased caspase-3 overexpression, and the prevention of podocyte and tubular cell apoptosis [106]. Again, in a rat model of diabetic-induced nephropathy, serial injections of MSC-EVs significantly improved renal functional parameters, including serum creatinine (Cr) and blood urea nitrogen (BUN), and slowed down the progression of fibrosis histologically [107]. These findings translated clinically as depicted by the first clinical trial, which investigated two doses of hUCMSC-derived EVs in 20 patients with stage III and IV CKD. The results were promising, demonstrating significant improvement in serum Cr and BUN in the experimental group compared to the placebo group. Additionally, participants treated with MSC-EVs demonstrated decreased serum concentration of proinflammatory TNF-a and increased concentration of anti-inflammatory TGF-b1 and IL-10, supporting the hypothesis of immune regulation in mediating improvement of renal function [108].

Another related but distinct clinical implication of MSC-EVs is their emerging role in solid organ transplant conditioning. In organ donation after cardiac death (DCD), organs are transplanted from a deceased donor to a living donor. During the transport process, organs are perfused with a solution to improve viability, but does not prevent ischemic injury. Compared to rat DCD kidney cold-perfused with standard protocols, those supplemented with solutions supplemented with MSC-EVs demonstrated upregulation of cell energy metabolism and ion membrane transport and decreased effluent material associated with global ischemic damage [109]. Taken together, supplementing ischemic organs in preparation for transplant with MSC-EVs may preserve viability and decrease reperfusion injury.

*4.4. Cartilage*

Osteoarthritis (OA) is the most prevalent chronic joint disease that irreversibly destroys the cartilage matrix. The precise molecular mechanisms involved in this degradation and the development of OA are poorly understood [110]. Current treatments are mainly directed toward relieving the symptoms of OA and include the use of nonsteroid anti-inflammatory drugs (NSAID) and pain medications [111]. While topical and oral NSAIDs have a moderate effect on pain relief, these therapies have many side effects and they cannot stop or reverse the ongoing cartilage degeneration [111,112]. The search for new beneficial interventions to decelerate the progression of OA or stop cartilage degeneration except for total joint replacement surgery is an imperative issue [110].

Studies demonstrating the chondroprotective and anti-inflammatory effects of MSC-EVs are particularly intriguing in the context of OA, the most prevalent rheumatic disease characterized by degradation of cartilage [113]. Historically, osteoarthritis is treated symptomatically with systemic anti-inflammatories or immobilization, or surgically through the removal of the affected joint or joint fusion as to eliminate the source of pain and disability. Promising results in clinical trials demonstrate MSCs ability to regenerate cartilage. A year following the intervention, there was no stem cell DNA detected in regenerative tissue, suggesting the role of MSCs as modulators with paracrine mechanisms, spurring further interest in MSC-EVs and sEVs [113,114].

Preclinical studies demonstrated that human and murine bone marrow-derived [115,116] and human embryonic stem cell-derived MSC-EVs and sEVs [117] promote cartilage regeneration. Human BM-MSC-EVs diminished the TNFa upregulation of COX2 when cocultured with OA chondrocytes and promoted cartilage regeneration [115]. In OA-like chondrocytes, murine BMMSCs reinstated the expression of chondrocyte markers and decreased catabolic and inflammatory markers in vitro, amounting to chondroprotective effects in vivo [116]. Zhang et al. published multiple studies utilizing human embryonic MSC sEVs in OA rat models. They found exosome CD-73 mediation of the AKT and ERK signaling pathways to manifest increased cellular proliferation and infiltration, matrix synthesis, and cartilage regeneration in murine knee and hip OA chondrocytes and distal femur osteochondral defects [117,118]. These findings were further elucidated in murine temporomandibular joint OA, in which there was exosome-mediated repair of matrix expression, subchondral bone architecture, and joint restoration, again through AKT and ERK signaling [119]. Intra-articular injection of ESC-MSC sEVs into surgically destabilized mouse knees decreased cartilage destruction and matrix degradation through an increase in type II collagen synthesis [120]. MSC-EVs have been shown to protect cartilage and bone degradation by increasing chondrocyte marker expression such as type II collagen and aggrecan, decreasing catabolic markers such as matrix metalloproteinase-13 (MMP-13) and ADAMTS5, decreasing inflammatory markers, and protecting chondrocytes from apoptosis [113]. Tofino-Vian et al. supported this finding by treating OA chondrocytes with AD-MSC-EVs, which showed a decrease in MMP activity and MMP-13 expression, while increasing cytokine IL-10 and collagen expression [121]. These regenerative and immunoregulatory properties make BM-MSC-EVs a favorable choice as an ideal therapy for OA. Given these promising preclinical results, the first clinical trial has been initiated, investigating osteochondral explants from arthroplasty patients treated with adipose-derived MSC-EVs in order to validate a cell-free approach in an ex vivo OA model (NCT04223622).

*4.5. Wound Healing*

Despite advances in wound management, almost 50% of chronic wounds are resistant to treatment [122]. MSC-EVs have anti-inflammatory, antiaging, and wound-healing effects, as shown in numerous studies performed in vitro and in vivo models [123]. EVs could have a massive effect on treatment of skin and wound healing. It is accepted that the basic mechanism of wound healing is inflammation.

The effect of menstrual blood-derived MSC-EVs on wound healing in diabetic mice caused a decrease in inflammation via induced macrophage polarization from a proinflammatory M1 to M2, an increase in neoangiogenesis through vascular endothelial growth factor A upregulation, and activation of the NF-kB signaling pathway [124]. In addition, the mice had a significant increase in wound closure and possibly less scar formation [124]. This finding was supported by the use of MSCs preconditioned with LPS in a cutaneous wound model in diabetic rats, which showed a well-timed transition from M1 to M2, with the resolution of chronic inflammation and curative effects on wound healing [125]. MSC-EVs can affect T cells, promoting an anti-inflammatory state. In a study by Du et al., peripheral blood mononuclear cells (PBMCs) were isolated from asthmatic patients and cultured with BM-MSCs [126]. The MSC-EVs showed an upregulation of IL-10 and TGF-beta 1 in PBMCs, leading to proliferation and an increase in the immunosuppression capacity of regulatory T cells, which play an important role in the development of asthma [126]. Wound-healing effects were seen in a diabetic foot ulcer model in rats where ADSC-EVs increased granulation tissue formation, blood vessel density, and growth factor expression, which accelerates cutaneous wound healing by decreasing the ulcerated area and relative fibrosis [127]. Topical administration of a gel containing MSC-EVs accelerated skin wound closure in rats [128]. This group went on to show that exposure to ADSC-MSC-EVs causes increased migration and proliferation of fibroblasts and keratinocytes, as well as activation of the AKT signaling pathway to help promote wound healing [128]. MSC-EVs have

been used to induce hair growth [129], help with epidermal barrier repair [130], and have beneficiary effects on the healing of irradiated wounds in mice [131].

In mammalian soft tissue, the physiologic process of wound healing occurs in three overlapping phases with the primary goal of reestablishing a functional skin barrier for protection from the environment. The first inflammatory phase involves hemostasis through activation of the clotting cascade, infiltration of monocyte-derived macrophages, and subsequent cytokine release to signal fibroblast infiltration to carry out the next phase. During the second proliferative phase, granulation tissue is formed via fibroblast creation of extracellular matrix that serves as the scaffold for re-epithelialization and endothelial cell mediation of angiogenesis. The final remodeling phase consists of collagen reorganization from type III to type I; wound contraction mediated by myofibroblasts. Wounds that are unable to progress normally through this dynamic process develop into chronic wounds if the cycle is inadequate, or excessive scars if the cycle is uncontrolled. Inflammation is involved in all phases of wound healing and has a direct effect on scar formation and pathology. Several preclinical studies investigated the role MSC-EVs and sEVs throughout the stages of wound healing to aid in the healing of chronic wounds and decreasing the formation of scar tissue.

In an in vitro model of wound healing, it was found that BM-MSC sEVs caused dose-dependent proliferation of fibroblasts in normal and chronic wounds and increased tube formation of endothelial cells through the activation of AKT, ERK, and STAT3 pathways and induction of growth factors [132]. Multiple studies focused on EVs and sEVs sourced from ADSCs. ADSC-derived sEVs demonstrated dose-dependent increases in fibroblast cell proliferation and migration as well as collagen synthesis via the PI3K/AKT pathway [128,133,134]. Further, for full-thickness incisions in a mouse model, ADSC sEVs expedited healing time and reduced scar formation [133,134]. The reduction in scar formation mediated by ADSC-derived sEVs may be through the regulation of collagen type III and I ratios and reduction of fibroblast differentiation into myofibroblasts [135]. Additionally, one study found a long noncoding RNA MALAT1 (metastasis-associated lung adenocarcinoma transcript-1) in ADSC-derived sEVs to be responsible for fibroblast cell migration in the context of ischemic wounds [136]. MSCs can affect the natural process of aging. Hu et al. looked at the antiaging efficacy of human dermal fibroblast (HDF) EVs derived from 3D spheroids in a mouse photo aging model [137]. An increase in procollagen type I expression, a decrease in MMP-1 expression via downregulation of TNF-alpha, and upregulation of TGF-beta were seen showing the anti-skin-aging properties of MSCs [137]. Nicotinamide adenine dinucleotide (NAD+) declines with age and plays a role in many important diseases of aging [138]. Yoshida et al. interestingly found that extracellular nicotinamide phosphoribosyltransferase (eNAMPT) is carried in EVs and enhances NAD+ biosynthesis [139]. eNAMPT-containing EVs were isolated from young mice and injected into aged mice, resulting in enhanced wheel-running activity and an extended life span in the older mice [139].

Investigation of pluripotent stem cell-derived MSC sEVs on wound sites in a rat model revealed that MSC sEVs expedited re-epithelization, decreased scar size, promoted collagen formation, and increased the formation and maturation of blood vessels in the wound. This was hypothesized to be due to the stimulation of HDFs and increase in collagen and elastin secretion in associated in vitro models [140]. Further, human umbilical cord MSC-derived exosome accelerates re-epithelization through WNT4 protein activation of ß-catenin signaling in a rat skin second-degree burn model [141]. The effect of sEVs and EVs in clinical wound healing is emerging, with the first clinical trials investigating nanoparticles from Wharton's Jelly MSC-conditioned media on chronic ulcerative wounds completing recruitment (NCT04134676) and another investigating the safety and efficacy of MSC-EVs in chronic dystrophic epidermolysis bullosa wounds (NCT04173650). Taken together, nanoparticles demonstrate significant promise in preclinical studies in their ability to optimize the characteristics of fibroblasts—the workhorses of the wound-healing cycle—to aid in the healing of chronic wounds and decrease scar burden.

## 5. EVs in Bioengineering Applications

Bioengineering is a diverse field that draws input from both the biological and engineering sciences to create new diagnostic and therapeutic modalities for patient care. Often, this relies heavily on advances in materials science and additive manufacturing technologies.

### 5.1. Materials

In tissue regeneration, the application of an appropriate engineering strategy for developing three-dimensional (3D) constructs possessing the ability to deliver bioactive agents at the site of action with adequate temporo–spatial control is crucial for restoring the structural and functional integrity of damaged tissues/organs. A number of natural polymeric materials, such as silk, collagen, fibrin, gelatin, and hyaluronic acid, have shown promise for fabrication of such constructs due to their structural versatility, low immunogenicity, biocompatibility, and biodegradability in physiological conditions [142]. Such polymeric hydrogels serve to retain the encapsulated drugs, particles, or cells for targeted therapeutic applications. Amongst them, encapsulated particles can further control the degradation of drugs while reducing cellular integration. Important parameters that are required to be optimized include cellular adhesion, material characteristics, surface topology, and charge. Moreover, porous scaffolds can enhance cell infiltration and proliferation. Thus, the combination of scaffold technology with EVs can greatly enhance their clinical applicability [143].

Hydrogels are a backbone in tissue engineering and represent a platform technology that has found wide clinical relevance, especially in reconstructive surgery and wound healing. However, hydrogels have also been used as an EV delivery system to mitigate the otherwise rapid clearance resulting from systemic administration. Mardpour et al. showed that such a strategy could be used to improve hepatic regeneration in chronic liver failure by sustaining MSC-EV release [144]. Unfortunately, hydrogels alone do not have the desired mechanical characteristics for all intended uses and often need to be combined with other materials. Silk also has longstanding clinical use, primarily as surgical suture. However, in order to improve scaffold physical characteristics, it has often been used as a composite material with other hydrogels. Now, it has been shown that these composite materials can further be loaded with EVs to impart specific therapeutic purposes. For example, silk composite hydrogels, when loaded with MSC-EVs, promoted wound healing and angiogenesis while reducing inflammation in a mouse wound-healing model [145]. While flat scaffolds are quite common for wound healing, silk composite scaffolds can also be shaped for more specific applications. Vorp et al. showed that when tubed scaffolds were used as aortic interposition grafts in rats, the EV-loaded cohort led to improved endothelium formation and smooth muscle proliferation, culminating in increased patency after eight weeks in vivo [146]. The continued development of manufacturing technologies will only serve to increase available options for scaffold shaping and biologic localization.

### 5.2. Bioprinting with EVs

Bioprinting is an enabling biofabrication technique that allows for the precise deposition of bioactives as per user-defined architectures and compositions [147]. Bioprinting methods have now been explored for developing scaffolds for delivery of EVs. For example, Diomede et al. and Pizzicannella et al. employed human gingival MSC-EVs complexes to develop 3D polylactide (PLA) scaffolds, which can release EVs at the desired levels [148,149]. To achieve the higher release of EVs, polyethylene imine was coated on PLA scaffolds. Subsequently, the authors demonstrated the positive effect of improving PLA-EV adhesion by this mechanism on bone regeneration. Higher osteoinductivity was confirmed by biocompatibility and differentiation assays as well as using next-generation sequencing for transcriptomic analysis. Further, in vivo investigations of these scaffolds on cortical calvaria defects of rodents have indicated enhanced osteogenicity and bone tissue growth (Figure 7A) [148]. In a further advancement to the study, 3D-bioprinted PLA supplemented with MSC-EVs has shown increased OPN, RUNX2, COL1A1, and

VEGF-A expressions in cells cultured on these scaffolds (Figure 7B) [149]. Additionally, the expression of miR-2861 and miR-210 was also found to be increased.

**Figure 7.** (**A**) Three-dimensional-printed poly(lactide) (PLA) scaffold and human gingival stem cell-derived (hGMSCs) extracellular vesicles (EVs). (i,ii) Topographic AFM image showing round morphology of EVs and polyethyleneimine (PEI)-engineered EVs (PEI-Evs); (iii) the images at low magnification showed construct integrated smoothly with the host; (iv) high-magnification images showing the new bone formation stained with acid fuchsin in both samples grafted at 6 weeks postsurgery (adapted and reproduced from [148]). (**B**) Three-dimensional-printed poly(lactide) (PLA)/Gingival Stem Cells/ Evs. (i) Representative μCT picture and SEM acquisition of 3D-printed PLA (3D-PLA); (ii) histological evaluation in vivo. After six weeks of grafting, samples were stained with von Kossa silver staining or Methylene blue and acid fuchsin images for the 3D-PLA/hGMSCs/Evs sample. Black arrows indicated blood vessels. Scale bar: 50 μm; (iii) 3D-MicroCT analysis showing 3D volume rendering and virtual transverse sectioning of 3D-PLA/hGMSCs/EVs (adapted and reproduced from [149]). (**C**) Three-dimensional coprinting of PCL with an alginate-based hydrogel encapsulating lyosecretome (i). (ii) SEM morphological and structural characterizations of 3D-printed PCL scaffolds loaded with lyosecretome. Presence (red arrows) or absence (blue arrows) of material deposition on PCL fibers is indicated. (iii) Scaffold geometry and dimensions of coprinting parallelepiped- and cylindrical-shaped scaffolds with a "soft heart" of lyosecretome-laden alginate (i.e., bioink) (adapted and reproduced from [150]). (**D**) Osteoblastogenesis by extracellular vesicle-associated bone morphogenetic protein 2. (i) Representative bright-field images of ALP assay (C2C12 cells) and mineralization assay (MC3T3 cells); (ii) ALP staining of C2C12s post-72 h seeding on bioprinted patterns with indicated 20 overprints (Ops); (iii) histological images showing H&E and Masson's trichrome staining of BMP2-EV-bioprinted implants (* indicates bone tissue); (iv) representative μCT 3D reconstructions of mouse leg scans containing either native EV or BMP2-EVs bioprinted implants. Arrow points to heterotopic ossification (adapted and reproduced from [151]).

Combination of 3D printing and freeze-dried formulation (lyosecretome) methods has been recently employed for fabrication of MSC secretome consisting of cytokines, proteins, and EVs. Two different strategies have been applied for fabrication of polycaprolactone (PCL)-based 3D-/printed scaffolds: in one approach, PCL was coprinted with MSC secretome incorporated in alginate hydrogels; and in another method, MSC secretome was adsorbed on prefabricated scaffolds. The absorption method allowed for a burst release of EVs from scaffolds, whereas slow release of EVs from encapsulating lyosecretome was observed in the coprinting method (Figure 7C). The kinetic studies confirmed the diffusion release of EVs from secretome from both loading methods. The release of EVs was also found to depend on concentration of polymer, crosslinking modes, scaffold geometries, etc. Such prototypes have been tested for bone tissue regeneration [150].

In another study, bone morphogenetic protein-2 (BMP-2)-loaded EVs were assessed through radiolabel-based assays [151]. Initially, it was observed that in vitro radiolabeled study of liquid-phase BMP2-loaded EVs demonstrated their cellular transport and osteogenic activities. While cell trafficking with free BMP2 confirmed the association of BMP2 within EV cargo is a natural process, in the natural BMP2-EV complex it was found that BMP2 is attached unstably on the EV surface. Therefore, a solid-phase bioprinted eBMP2-EV niche was developed on collagen-rich matrix for both in vitro and in vivo assessment (Figure 7D). Studies showed that EV incorporated with BMP-2 induced in vitro and in vivo osteogenic differentiation in picogram and nanogram level doses, respectively. The study suggested the role of BMP-2-based cell signaling for internalization of BMP-2-loaded EVs, which differ from the established growth factor-based mechanism. EVs isolated from human umbilical vein endothelial cells can be used as bioadditives for developing unique bioinks for developing 3D-bioprinting constructs.

Gelatin methacrylamide (GelMA) bioink has also been used for fabricating bioprinted EV-based scaffolds. In this process, gelatin was reacted with methacrylic anhydride and bioinks were separately loaded in EVs for different culture conditions. Bioinks alginate and calcium chloride crosslinkers were allowed to flow through a single-syringe pump at different flow rates. Multilayer scaffolds with aligned fibers were fabricated by keeping 50 μm distance in between. Several parameters of this bioprinting model were studied, such as the effect of EVs on the bioink viscosity, concentration of the crosslinker on bioprintability, scaffolds with different fiber orientations, fiber diameters on bioprinting speed, etc., for developing hierarchical vasculature constructs. Further, the distribution of EVs on the bioprinted matrix was tested at different magnifications. Higher-magnification images showed remarkable differences in surface properties where matrices with EVs showed rough surface topology while EV-free matrices consisted of smooth surface characteristics. The integrity of the EV membrane was investigated through isolating EVs from scaffolds through enzymatic digestion. Results showed that the compatibility of bioprinting with EV encapsulation as the EV membrane was not damaged during bioprinting and the size of the EV remained unchanged. Such EV-based bioinks show immense potential for supporting the transplantation and revascularization of ischemic regions, and can also meet the challenge of microvessel integration in tissue-engineered constructs. However, in-depth standardization is still required for clinical-grade EV production [152]. In vivo implantation of such 3D-bioprinted scaffolds supported the in situ vascularization through the formation of microvessels. Results indicate the possibilities of therapeutic application of this technology in critical clinical needs, especially for ischemic tissue revascularization (Figure 8A) [152]. Towards exosome-based therapeutics, Chen et al. used desktop-stereolithography technology to fabricate a 3D-printed cartilage extracellular matrix (ECM)/GelMA/exosome scaffold with radially oriented channels. The results showed the system restored cartilage mitochondrial dysfunction, enhanced chondrocyte migration, and macrophage M2 polarization along with osteochondral defect repair in a rabbit model (Figure 8B) [153]. To overcome the limit of EV production, a recent study reported a scalable method for high-quantity EV manufacturing. Herein, MSCs were densely encapsulated in micrometer-scale hydrogel fibers by coaxial bioprinting, and the results indicated that

the developed system augmented particles by ~1009-fold compared to conventional 2D culture, while preserving their proangiogenic properties (Figure 8C) [154]. Addressing scalable EV production, bioreactor systems have also been successfully used. Patel et al. reported a 3D-printed scaffold-perfusion bioreactor system with enhanced response of dynamic culture on EV production from endothelial cells (ECs) [155].

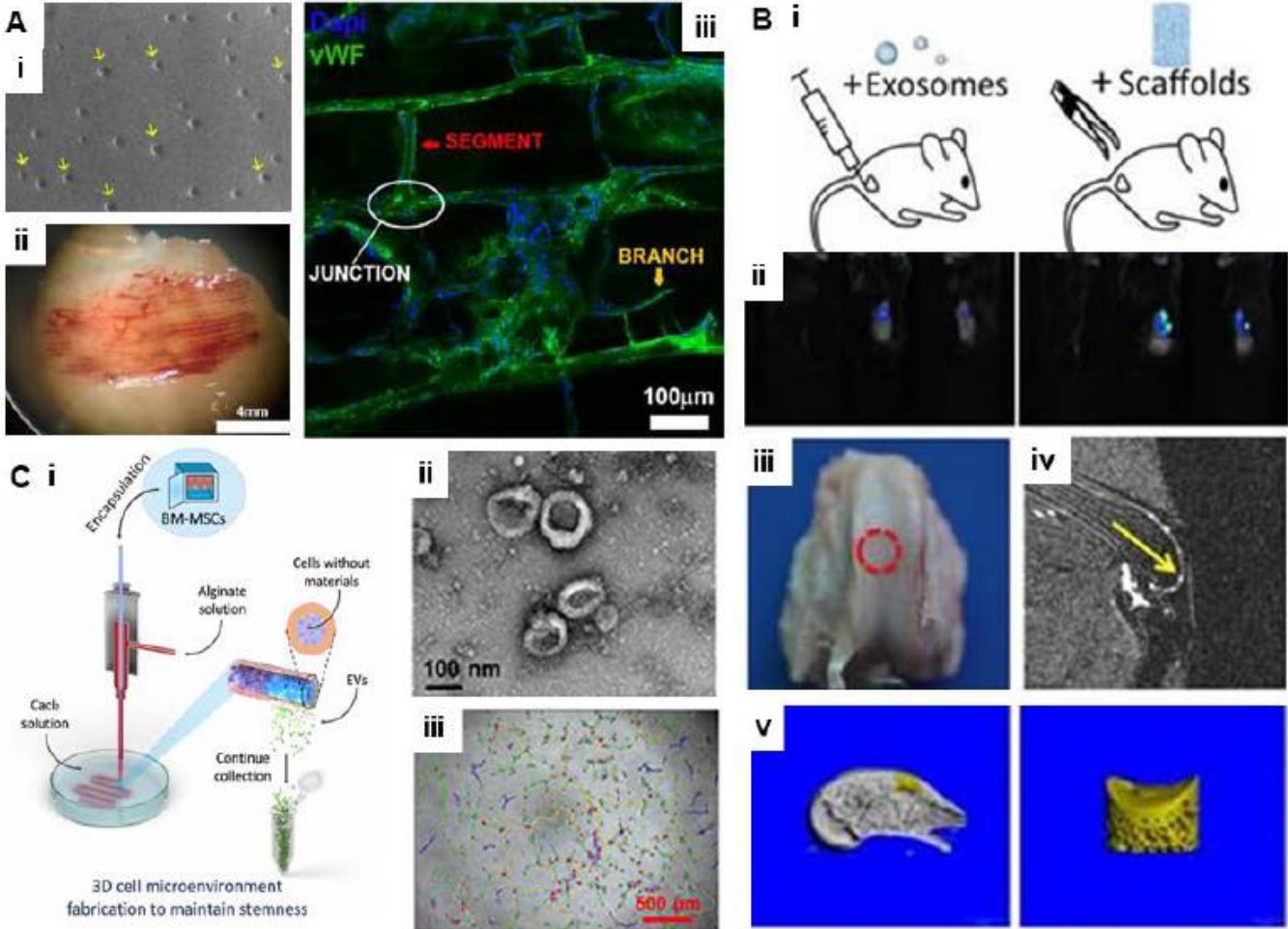

**Figure 8.** (**A**) Three-dimensional-bioprinted endothelial-derived extracellular vesicles. (i) SEM images of 3D-bioprinted hydrogels (GelMA with EVs). Yellow arrows show vesicles; (ii) representative images of serum-free M200 medium in hypoxia (SM Hypoxia) EV sample explanted neovascular networks after EDTA decalcification treatment; (iii) immunofluorescence image of vWF-positive vessels showing branching and angiogenesis (adapted and reproduced from [152]). (**B**) Three-dimensional-printed extracellular matrix/mesenchymal stem cell sEV scaffold for osteochondral defect repair. (i) In vivo fluorescence imaging of sEVs in PBS and in the 3D-printed scaffold. Both groups of sEVs were labeled with Vybrant DiO (green) dye before implantation; (ii) macroscopic evaluation of the osteochondral defect regions at 12 weeks; (iii) MRI imaging of repaired knees at 12 weeks after surgery (yellow arrow indicates the repaired sites); (iv and v) μCT reconstruction imaging of repaired knees at 12 weeks after surgery (adapted and reproduced from [153]). (**C**) Three-dimensional coaxial bioprinting for EV production. (i) Scalable production of highly enriched EVs using 3D cell fiber culture. MSCs and alginate solution were extruded through the inner and outer streams of the coaxial printer head, respectively, to form a dense 3D cell fiber that promotes cell survival and stemness maintenance; (ii) TEM of EVs isolated from 3D cell fiber culture showing a typical 'cup-shaped' morphology; (iii) evaluation of endothelial cell tube formation with EVs using a Matrigel matrix. Morphology of the endothelial network stimulated with 3D ($2.8 \times 10^9$ particles mL$^{-1}$) EVs (adapted and reproduced from [154]).

The microfluidic-based "deterministic lateral displacement (DLD)" technique has been widely developed through 3D-printing or additive-manufacturing processes, in which arrays of micro/nanoposts are incorporated in microfluidic channels. Such systems facilitate the transport of suspended particles, and flow parameters can be modulated by altering the variables of geometric design. A "two-photon "direct laser writing (DLW)" method has been shown to be promising for manufacturing of DLD arrays, which can efficiently transport the submicron particles. The experimental result on testing of 860 nm fluorescent particles using DLD arrays fabricated by DLW-printing aided in understanding the hydrodynamic barrier of fluorescent particles along the angled arrayed of microposts. The lateral displacement of the study noted $15.3 \pm 8.6$ µm over a 500 µm channel length. Results indicate the promising application of 3D-printed DLD systems for the manipulation of submicron particles [156] that can be used for efficient bioprinting of EVs.

## 6. Clinical Translation

MSC-EVs are highly versatile and potent biological structures in terms of regenerative and anti-inflammatory capabilities for the creation of effective cell-free therapeutics. They exhibit targetable biodistribution, improved stability and handling, and decreased toxicity with no thrombogenic or tumorigenic hazards following intravascular delivery. These properties allow for their effective clinical applications. Huang et al. investigated the differences in protein from MSC-EVs and suggested that the isolation methods might introduce variations in the protein composition in sEVs, which directly affects their biological function [157]. Tieu et al. reviewed and evaluated in vivo preclinical studies using MSC-derived EVs as an intervention for any animal disease model [158]. The results indicated that despite the variations, 81% of studies indicated benefits for all evaluated outcomes, and 97% of studies reported improvements for at least one outcome. In 72% of investigations, EVs' involvement improved outcomes, organ function, and survival compared to unmodified MSC-EVs. Additionally, advantages of this cell-free therapy are seen in diseases affecting all organ systems with fundamentally different pathophysiology, implying that MSC-EVs may be used to treat a range of disorders with little risk of side effects. To eliminate any source of variability and animal-related contaminations, the use of xeno- and EV-free culture medium is advised for clinical applications.

To scale up EV production, 3D culture in bioreactors, such as multilayered cell culture flasks, hollow fiber bioreactors, stirred-tank bioreactors, and spheroidal aggregates of MSCs has been investigated. Particularly, the stirred-tank and hollow fiber bioreactors, which are closed, scalable, and offer a high surface-to-volume ratio for MSC growth, have shown promising results [159,160]. The results have demonstrated that EV production in bioreactors is at least 40 times higher than in 2D culture methods [161]. For good therapeutic effect and for toxicity, biodistribution, or pharmacokinetic studies, highly purified and homogenous EV fractions are a requisite that majorly depend on the isolation method. An ideal isolation technique should be scalable, cost-effective, compatible with a high-throughput production process, and preferably be a closed system. Several methods have been investigated over time. The technique of ultracentrifugation (UC) has traditionally been regarded as the most common method for isolating MSC-EV from either cells or biological fluids. However, associated drawbacks with UC include impairing the integrity of EVs, being relatively time-consuming, necessitating processing samples in large volumes, and requiring specialized lab equipment. Nevertheless, sequential centrifugation steps have been used for the large-scale production of clinical-grade MSC-EVs [160]. Further, the impurities and other protein contaminants in the isolated materials make it challenging to determine whether the biological activity is due to EVs or the co-isolated soluble mediators. Monguió-Tortajada et al. used size-exclusion chromatography (SEC) for isolating well-defined EVs similar to those of parental MSCs, highlighting the importance of well-purified preparations of MSC EVs to achieve immunosuppressive effects [162]. Additionally, other techniques involving affinity chromatography solutions (heparin affinity binding [163] or shiga or cholera toxin binding [164]) may be appropriate for purely analytical purposes;

however, these may not entirely meet the current good manufacturing practice (GMP) regulations and the demand for large-scale purification.

For the industrial and clinical use of EVs, it is necessary to implement the regulatory issues for EV production and clinical uses. Lerner et al. in an ISEV (International Society for Extracellular Vesicles) position paper explored how to categorize EVs according to the anticipated active ingredients in biological medicines and pharmaceuticals along with discussing the legal problems in EV-based therapeutics [165]. Compliance with the regulatory frameworks is pivotal for the approval of EV-based therapies and their large-scale implementation. In compliance, several companies have already developed various EV-based therapies for a range of diseases and therapeutic targets, and MSC-EVs account for about 40% of these products [166]. Large-scale processes are being implemented using selectively closed systems for improved safety to assure reliable production procedures. These processes are mostly bioreactor technologies for EV production and ultrafiltration technologies for EV purification. For the manufacturing of MSC-EV or secretome products, several GMP-compliant procedures have been created [167,168]. The key problem associated with the industrialization of EV-based medicines for regenerative medicine is to develop novel manufacturing techniques under GMP for EV scalable production and isolation. Different production methods will generate different products; thus, it is essential to develop a standardized operating procedure (SOPs) using standardized and reproducible assays to produce a defined and stable EV product. Overall, although most of the preclinical trials involving MSC-EV-based therapeutics are yet to be translated into clinical trials, the results from several studies are promising for future clinical applications.

## 7. Conclusions

EV-based therapies possess immense potential in the field of regenerative medicine in the future because of their bioactive cargo and therapeutic potential for a variety of diseases. Existing studies provide solid evidence that EVs generated by MSCs of different origins demonstrate paracrine activity like that of their parental cells. However, several challenges must be considered and overcome before their clinical utility is achieved, including isolation and purification producing homogenous EVs; scaling up of EV production, optimizing, and setup of GMP-compliant procedures; standardized SOPs for better reproducibility; and optimizing the safety, immunogenicity, or optimal doses of EVs. Despite the challenges, based on the growing amount of evidence, it could be said that when refined and thoroughly characterized in terms of their bioactive cargo and mechanisms of action, MSC-EVs will be a suitable alternative to cell-based therapies as allogenic treatments for regenerative medicine applications.

**Author Contributions:** Major contribution towards: writing manuscript: C.M.; writing manuscript on bioprinting: P.D.; revised manuscript, Figures 7 and 8, bioprinting section: Y.P.S.; writing manuscript: A.L.; editing manuscript: S.H.; revised whole manuscript, rewriting, editing: I.A.E.; writing and correcting bioprinting section: I.T.O.; writing and editing manuscript: D.J.R.; Designed review article, wrote manuscript, preparing figures and supervised: S.V.K. All authors have read and agreed to the published version of the manuscript.

**Funding:** Department of Surgery at Penn State College of Medicine, Hershey PA 17033 USA.

**Institutional Review Board Statement:** Not applicable.

**Informed Consent Statement:** Not applicable.

**Data Availability Statement:** Not applicable.

**Acknowledgments:** We acknowledge financial support by the Department of Surgery at Penn State College of Medicine.

**Conflicts of Interest:** The authors declare no conflict of interests.

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
