# Peer review of "Mesenchymal Stem Cell-Derived Extracellular Vesicles for Therapeutic Use and in Bioengineering Applications"

_cells, doi:10.3390/cells11213366_

Round 1

Reviewer 1 Report (Previous Reviewer 1)

Comment 1. The authors should improve the quality of the images

Comment 2. The exosome should be changed by sEV

Author Response

  1. Atuhors should improve quality of images.

Response: As reviwer suggestion, we improved image quality in figures

  1. The exosomes whould be changed sEVs

Response: We agree with reviewer as recent change in namenclature, we changed all exosomes to sEVs in manuscript

Reviewer 2 Report (Previous Reviewer 2)

The present review titled " Mesenchymal Stem Cell-Derived Extracellular Vesicles for Therapeutic Use and in Bioengineering Applications"appears not particularly interesting for the following reasons:

1-the general shape reflects more a list that an elaborated view. All the information are already present in the literature and there is not novelty.

2-figure 2 is not clear: why it is written stem cell?

3-paragraph 2: it is more a list of finding than a critical correlation. 

4- for the figure 3 it is not clear the immunomodulatory effect that is mentioned in the text of the review. 

5-figure 4 is very general and all the organs are not described in the text.

6-in general the acronyms should be checked. EV-MSC-hUCs or hUC-MSC-EVs? 

7-par 2.5: more text should be added and better explained and , as for other parts, also here references are lacking.

8- in some parts the text is repetitive and phrases like " MSCs-derived EVs have therapeutic potential to treat inflammatory and other diseases (Figure 6) [82]" are without meaning. Which other diseases?

9- part 5 is particularly difficult to read: a lot of repetition in the different paragraphs, lack of references and some text is really not fitting with the rest.

Author Response

  1. Figure 2 is not clear, why it it written stem cell

Response: we update image and written as MSCs

  1. Figure 3, it is not clear the immunomodulary effect that mentioned

Response: we updated text and removed immunomodulatory as suggested.

  1. Figure 4 general and all the organs are not described in the text

Response: we updated figure, described organs literature in review manuscript changed lines to red and non-described remain in black color, as limited literature on those organs.

  1. In general the acronyms should be checked for EV-MSC-hUCs or hUC-MSC-EVs

Response: we appreciate reviewer, we updated manuscript as suggested.

  1. In some parts the text is repetitiave like MSCs-dervied EVs have therapeutic poential to treat inflammatory and other diseases (Figure 6)

Response: we udpated as needed in manuscript and also figure 6 legend

  1. Part 5 is particulary difficult to read

Response: we updated as suggested by reviewer.

Round 2

Reviewer 2 Report (Previous Reviewer 2)

Figure 5 is of very low resolution. 

This manuscript is a resubmission of an earlier submission. The following is a list of the peer review reports and author responses from that submission.

Round 1

Reviewer 1 Report

Mesenchymal Stem Cell Derived Extracellular Vesicles for 2 Therapeutic Use and in Bioengineering Applications

cells-1642925

 Comment 1. The authors should be used the nomenclature small extracellular vesicles and large extracellular vesicles in the manuscript following MISEV 2018.

Comment 2. In the Figure 1, the size of exosomes (small extracellular vesicles) is not the same than the line 33 in the manuscript.

Comment 3. The quality of the images should be improved.

Comment 4. In the line 40, the cytokines, chemokines, growth factors, interleukins, transcription factors should include inside proteins because in the manuscript they appear independly.

Comment 5. The authors should in the lines 51-53, indicate in which disease the EVs from MSCs could use them and why?

Comment 6. In my opinion, the section 1.1, it is not necessary because the manuscript is based on mesenchymal stem cells.

Comment 7. The authors should be re-written the lines 78-80, because the sentences is an affirmation and in the end is a hypothesis.

Comment 8. The section 2.4 should be separated in several sections because in this part, the authors reviewed several aspects in cancer, neurodegenerative diseases and muscle. It is really complicated following the story in this part.

Reviewer 2 Report

The review by McLaughlin and colleagues is on extracellular vesicles. Different cell sources, clinical applications and one paragraph on bioprinting, EVs as biomarkers: all these points are described as other reviews already present in literature. This manuscript is original but is very similar to other reviews that have the same structures. The first figure is very similar to other papers on EVs. Not original considerations or specific point of view are written. For this reason, in the present form, this review is not more useful in respect to what is already present in the literature.